# UWGAN: Underwater GAN for Real-world Underwater Color Restoration and Dehazing

## Abstract

In real-world underwater environment, exploration of seabed resources, underwater archaeology, and underwater fishing rely on a variety of sensors, vision sensor is the most important one due to its high information content, non-intrusive, and passive nature. However, wavelength-dependent light attenuation and back-scattering result in color distortion and haze effect, which degrade the visibility of images. To address this problem, firstly, we proposed an unsupervised generative adversarial network (GAN) for generating realistic underwater images (color distortion and haze effect) from in-air image and depth map pairs based on improved underwater imaging model. Secondly, U-Net, which is trained efficiently using synthetic underwater dataset, is adopted for color restoration and dehazing. Our model directly reconstructs underwater clear images using end-to-end autoencoder networks, while maintaining scene content structural similarity. The results obtained by our method were compared with existing methods qualitatively and quantitatively. Experimental results obtained by the proposed model demonstrate well performance on open real-world underwater datasets, and the processing speed can reach up to 125FPS running on one NVIDIA 1060 GPU.

## 1 Introduction

In recent years, underwater vision plays an important role in a lot of different applications. Therefore, underwater image processing has received extensive attention and research due to the poor underwater imaging environment and image quality. The main reason is the scattering and attenuation of light, the scattering results in haze effect, and the attenuation of light leads to color cast.

So far many image enhancement algorithms have been proposed, such as white balance algorithm (Liu Y C, 1995), gray world algorithm (Rizzi A, 2002), histogram equalization (Pizer S M, 1987) and fusion algorithm (Ancuti C, 2012), however, these methods are not based on the underwater physical imaging model, so it is challenging and ineffective to apply these algorithms to different underwater scenes directly.

Many underwater image enhancement algorithms based on imaging models have been proposed. For instance, He et al (He K, 2010) proposed a dark channel prior (Dark channel prior, DCP) dehazing algorithm based on many experiments. Chiang et al (Chiang J Y, 2011) apply DCP model on underwater image dehazing problem. These traditional methods are not intelligent, it is very time-consuming to calculate the characteristics of the image.

In these years, the deep learning network developed rapidly, especially the convolutional neural network (CNN), which is used in image classification (Krizhevsky A, 2012), object detection (Redmon J, 2016), and motion recognition (Kuehne H, 2011), the performance is much better than traditional methods. However, the current research on underwater image enhancement using CNN is limited due to lack of underwater datasets. It is difficult to obtain images without water in real-world underwater scenes. Therefore, using synthetic underwater datasets is an important approach (Anwar S, 2018; Ancuti C, 2016; Uplavikar P, 2019). Some model based on generative adversarial network (GAN (Goodfellow I, 2014)) are used to generating realistic underwater images. For instance, CycleGAN (Zhu J Y, 2017) generates images through

style transfer. WaterGAN (Li J, 2017) takes in-air images, depth maps and noise vectors as input, followed by a camera model, then output synthetic images. Based on our experimental results, the image generated by WaterGAN suffers color noise and they differ a lot from real world underwater images. Therefore, to generate realistic underwater images with both color cast and haze effect, we improved the underwater imaging model, and proposed an unsupervised GAN based on this model to generate realistic underwater images from clear in-air images. Then, U-Net with different loss functions (Ronneberger O, 2015) is trained to enhance underwater images through synthetic datasets. Finally, the performance of the proposed algorithm is validated on real underwater images as well as underwater target detection datasets for both low-level and high-level computer vision tasks. The experimental results show that the proposed method can recover the underwater image while maintaining structural similarities. Apart from this, the effects of different loss functions in U-net are compared, the most suitable loss function for underwater image restoration is suggested based on the comparison (This part can be found in APPENDIX), which provides a new idea for underwater image enhancement.

## 2 OUR PROPOSED METHOD

To generate the realistic underwater images (color casts, low contrast and haze effect), we improved underwater imaging model, and proposed an underwater generative adversarial network (UWGAN), which takes in-air RGB-D images and a sample set of underwater images of a specific survey site as input to train a generative network adversarially. These synthetic underwater images, which were used to train a restoration network based on U-Net (Ronneberger O, 2015) that can enhance underwater images in real-time.

### 2.1 IMPROVED UNDERWATER IMAGING MODEL

As is well known, a simplified underwater imaging model is shown in Eq. 1.

$$I(x) = J(x)T(x) + A(1 - T(x))$$
$$T(x) = e^{-\beta(\lambda)d(x)} \quad\quad\quad\quad (\text{Eq. 1})$$

where, $I(x)$ is the light intensity of each pixel $x$. $J(x)$ is the initial irradiance that not propagating through the water. $T(x)$ is the transmission map of the scene. $A$ is the atmospheric ambient light of the scene. $\beta$ is attenuation coefficient of light of different wavelengths $\lambda$, and $d(x)$ is the range between the scene and the camera.

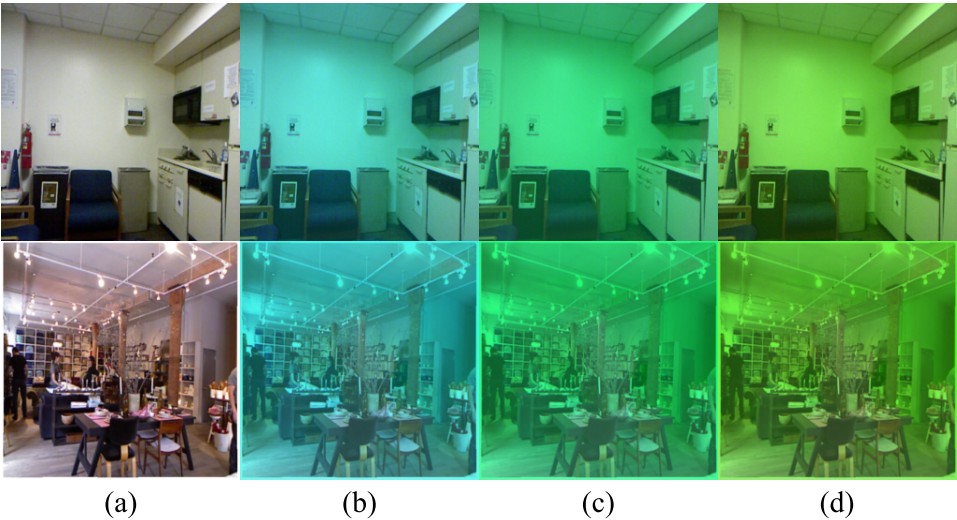

(a)        (b)        (c)        (d)

Figure 1. Synthetic underwater-style images through Eq. 2. (a) are in-air sample images, (b)-(d) are synthetic underwater-style sample images of different water types.

We can generate underwater-style images using the in-air image and its depth map by Eq. 1, which can well simulate color cast caused by light attenuation in water. However, it is difficult to simulate the haze effect caused by the scattering of water impurities. As shown in Figure 4, obvious haze effect can be observed on real underwater images. Inspired by related dehazing methods (Ancuti C, 2016), we improved the second term in Eq. 1. The improved imaging model is shown in Eq. 2.

$$I(x) = J(x)T(x) + AT(x)\big(1 - T'(x)\big)$$
$$T'(x) = e^{-\alpha d(x)} \tag{Eq. 2}$$

where, $AT(x)$ is ambient light based on the light attenuation of different wavelength. $\alpha$ is the scene scattering coefficient, which corresponds to the scattering coefficient in the atmospheric imaging model, and $\alpha$ is set by default to 1, corresponding to a moderate and homogeneous haze effect. Three types of realistic underwater images were synthesized with color cast and haze effect are shown in Figure 1.

## 2.2 UWGAN FOR GENERATING REALISTIC UNDERWATER IMAGES

Underwater-style images are generated based on Eq. 2, whose parameters are estimated through adversarial learning using GAN, as shown in Figure 2.

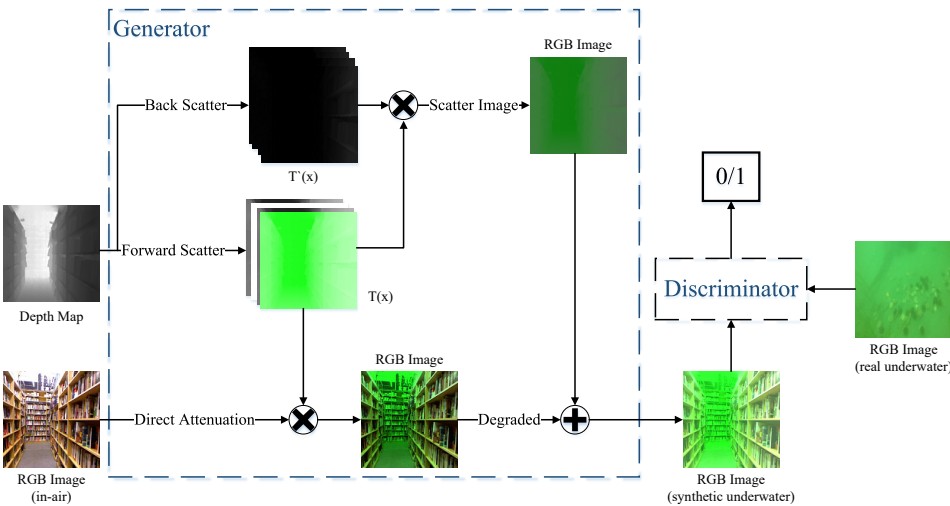

Figure 2: UnderwaterGAN architecture. UWGAN takes color image and its depth map as input, then it synthesizes underwater realistic images based on underwater optical imaging model by learning parameters through generative adversarial training.

## 2.3 UNDERWATER IMAGE RESTORATION BASED ON U-NET

U-Net is used for color restoration and haze removal of underwater images. A detailed description of U-Net architecture proposed in the paper is shown in Figure 3. Firstly, a degraded underwater RGB image is resized to 256x256 and then fed into the encoder part of U-net. In the encoder, the image is finally downsampled into a 32x32x256-dimensional latent vector through a series of convolution and max-pooling operations. In each downsampling stage, 3x3 convolution with a stride of 1 followed by a rectified linear unit (ReLU) activation function are conducted twice, then a 2x2 max pooling with a stride of 2 is used. The number of feature maps are doubled at each stage. In the decoder part, upsampling is done from the latent high dimensional vector back to the original input size sequentially. After each upsampling operation, output tensor is concatenated to the corresponding symmetric layer in the encoder side, then followed by two consecutive convolution layers and a rectified linear activation layer. The number of feature maps is gradually reduced to three channels.

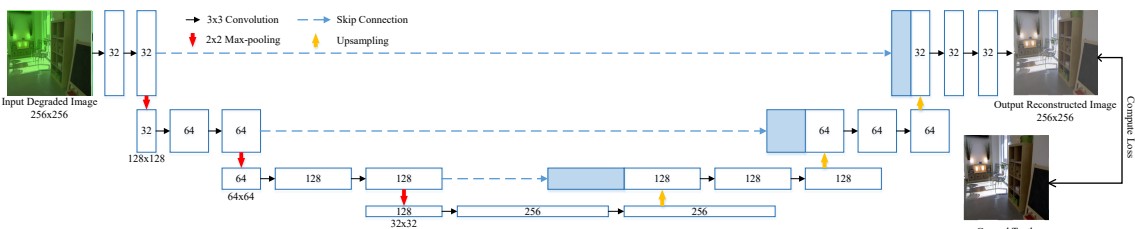

Figure 3: Proposed U-net Architecture for underwater image restoration and enhancement.

## 2.4 DATASET

The in-air datasets we used are images of indoor scenes that has been labeled in the NYU Depth dataset V1 (Silberman N, 2011) and V2 (Silberman N, 2012), which contain a total of 3733 RGB images and corresponding depth maps. The underwater dataset contains real-world underwater images collected from marine organisms' farms (including scallops, sea cucumbers, sea urchins, etc.), which can be roughly divided into two categories, one contains near-field green hued images (RealA), and the other contains blue-green hued images of far-field scenes (RealB). We also use underwater open datasets (Li C, 2019) (RealC) as testing sets, where RealA contains 2069 underwater images, RealB contains 2173 underwater images, and RealC contains 890 underwater images. Several typical images of the datasets are shown in Figure 4.

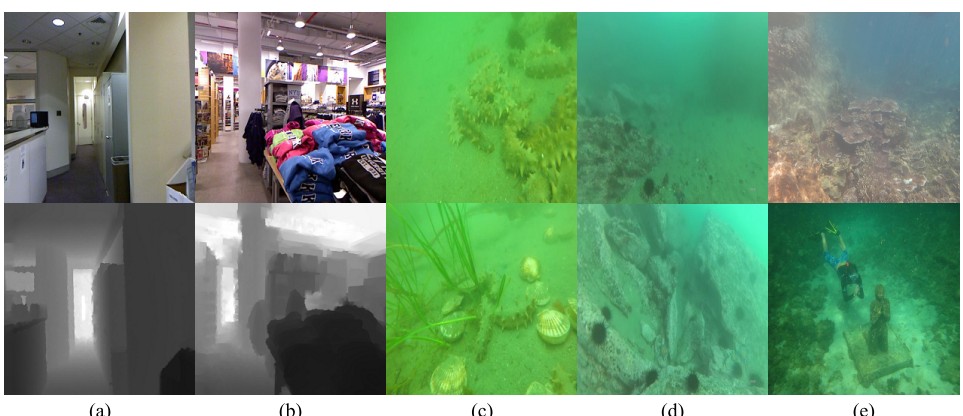

(a)   (b)   (c)   (d)   (e)

Figure 4: Typical images of datasets. (a)-(b) are color images and depth maps of NYU-Depth datasets, (c) are sample images of RealA dataset, (d) are sample images of RealB dataset, (e) are sample images of RealC dataset.

## 3 EXPERIMENTAL SETUP

The training settings of our proposed method are presented in details in this section. Our models are trained in the computer with the following configurations: Intel i7 HQ 8700 processor, 16GB RAM, NVIDIA TITAN X 12GB graphics card.

Firstly, UWGAN is trained to synthesize underwater-style images using the NYU-Depth Dataset, RealA and RealB datasets. Our model was trained for 30 epochs, using Adam optimizer with a learning rate of 0.0001, and the momentum term was set to 0.5. The batch size was set to 64 with output images set to 256x256. Secondly, U-net is trained as an image enhancement network using synthetic pairs. The batch size was set to 32 and the output image size is 256x256. The learning rate is set to 0.0001 according to Adam optimizer, our model is trained for 200 epochs.

## 4 RESULT AND DISCUSSION

In this section, we quantitatively and qualitatively compare our proposed method with several representative underwater image enhancement algorithms, including Unsupervised Color Correction Method (UCM) (Iqbal K, 2010), Histogram equalization (HE) (Hummel R, 1975), Multi-Scale Retinex with Color Restoration (MSRCR) (Rahman Z, 1996), Fusion (Ancuti C, 2012), Underwater Dark Channel Prior (UDCP) (Drews P, 2013), Image Blurriness and Light Absorption (IBLA) (Peng Y T, 2017), Underwater Color Correction using GAN (UGAN) (Fabbri C, 2018), WaterGAN-color-correction (WaterGAN) (Li J, 2017).

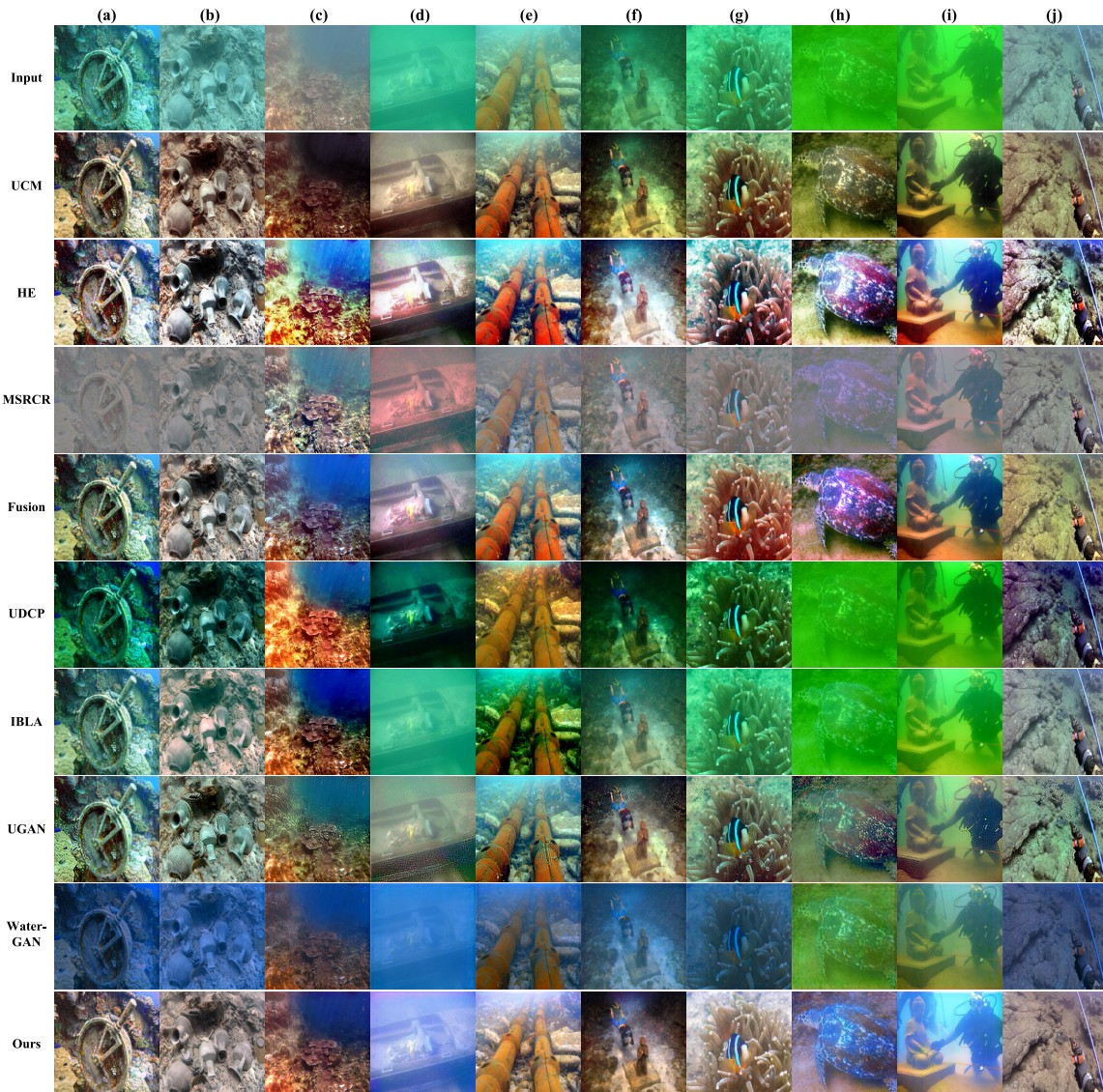

Figure 5: Qualitative comparisons for samples from the real-world underwater image dataset RealC. (a)-(j) represent the samples selected from RealC.

We employ a non-reference metric, UIQM (Panetta K, 2015), for the quantitative assessment of underwater image quality on RealA, RealB, and RealC datasets as no ground truth scenes are available as the reference for real-world underwater images. Besides, we employ three full-reference metrics, namely MSE, PSNR (Hore A, 2010), SSIM, for assessment image quality on synthetic datasets. To reasonably assess the time spent on various algorithms, we resize all images to 256x256, which provides a stable output for enhancements in later experiments.

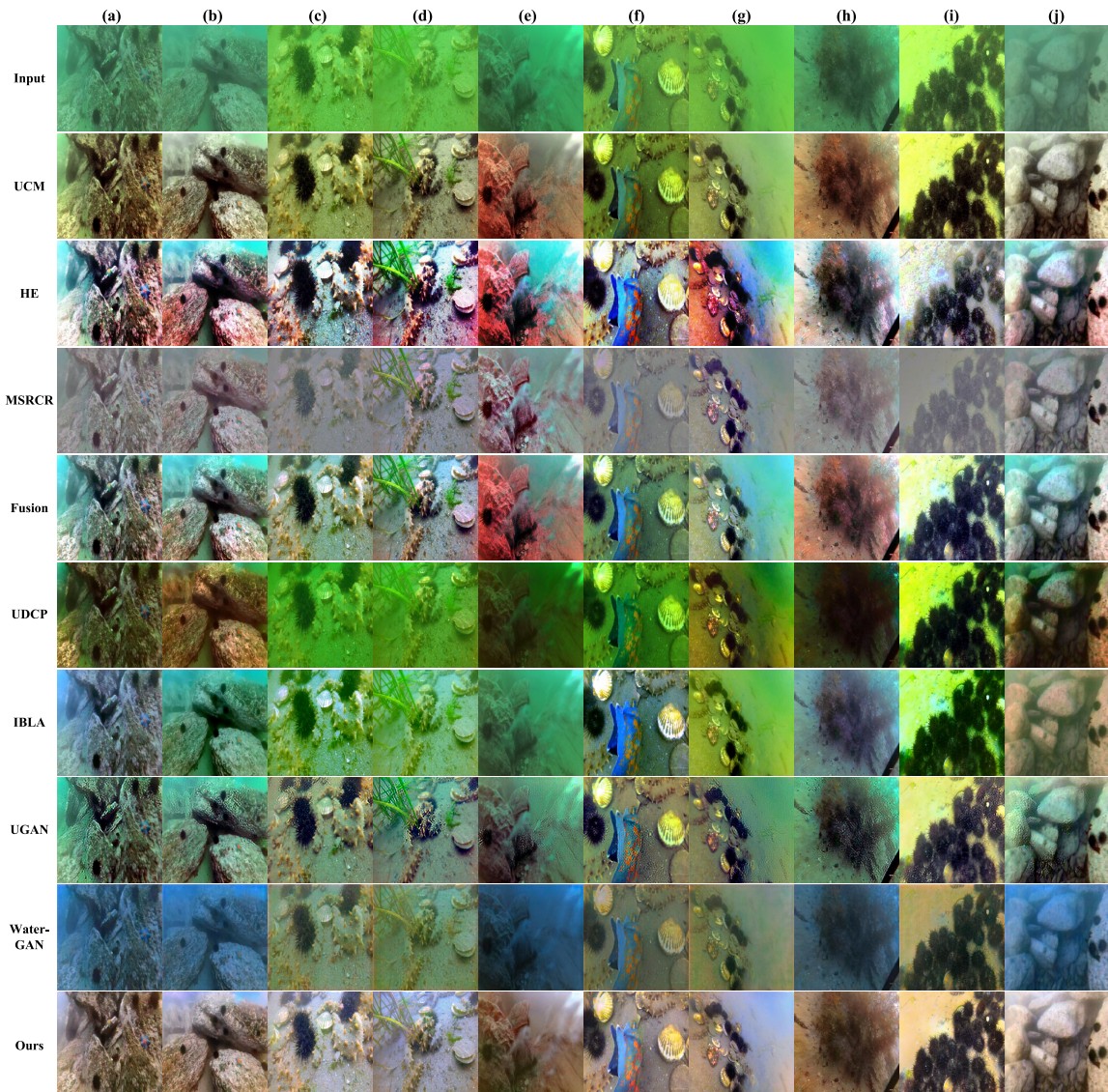

Figure 6: Qualitative comparisons for samples from real-world underwater image dataset RealA and RealB. (a)-(j) represents the samples selected from RealA and RealB.

Firstly, we compare the capabilities of different methods to improve the image visibility on the RealA, RealB, and RealC datasets. The qualitative comparison is shown in Figure 5 and 6. Most methods can improve the quality of images of a slight haze effect. UCM, HE, and Fusion can enhance the brightness and contrast of the image, but are less uniform for color restoration and seem to be over-enhanced in some areas of the image. The results of MSRCR appear to have a suitable hue but lack sufficient saturation and contrast. UDCP and IBLA do not recover well for green-toned images, they make the image darker but enhance the contrast of the image. UGAN, WaterGAN can enhance the contrast of the image but they don't recover color well and generate some artifacts, which destroy the structural information of the image. The proposed method can recover the color of degraded underwater images while keeping a proper brightness and contrast.

Table 1 and Table 2 quantitatively show the scores of sample images in Figure 5 and Figure 6 respectively. Our proposed method has achieved the highest scores in (a), (c) and (f). In addition, the average quantized scores evaluated on RealA, RealB, and RealC datasets are shown in Table 3. Our model achieves the best

scores in terms of color restoration.

Table 1: Quantitative UIQM values of samples in Figure 5. The greater the UIQM values, the better the enhanced results, with blue representing the maximum and green representing the minimum.

| Assessments | Methods | (a) | (b) | (c) | (d) | (e) | (f) | (g) | (h) | (i) | (j) |
|---|---|---|---|---|---|---|---|---|---|---|---|
| | Input | 5.475 | 4.995 | 4.171 | 3.523 | 4.554 | 4.842 | 4.868 | 4.459 | 4.195 | 4.619 |
| | UCM | 5.253 | 5.320 | 5.200 | 4.447 | 4.870 | 5.219 | 5.181 | 5.130 | 4.672 | 5.158 |
| | HE | 5.080 | 5.369 | 4.814 | 4.779 | 4.907 | 4.925 | 5.174 | 5.215 | 4.493 | 5.247 |
| | MSRCR | 4.047 | 4.636 | 5.229 | 4.135 | 4.516 | 4.528 | 4.465 | 4.259 | 4.022 | 4.684 |
| UIQM | Fusion | 5.329 | 5.460 | 5.095 | 4.546 | 4.970 | 5.181 | 5.295 | 5.220 | 4.544 | 5.145 |
| | UDCP | 4.820 | 4.704 | 4.727 | 4.836 | 5.255 | 4.440 | 4.435 | 3.830 | 3.757 | 5.385 |
| | IBLA | 5.468 | 5.302 | 3.867 | 3.559 | 20.606 | 4.861 | 4.941 | 3.537 | 3.659 | 4.999 |
| | UGAN | 5.326 | 5.287 | 5.325 | 4.204 | 4.846 | 5.022 | 5.126 | 4.947 | 4.353 | 5.122 |
| | WaterGAN | 5.024 | 4.934 | 4.833 | 2.763 | 4.594 | 4.414 | 4.547 | 4.879 | 3.953 | 4.700 |
| | Ours | 5.602 | 5.387 | 5.379 | 4.219 | 4.820 | 5.327 | 4.868 | 5.110 | 3.922 | 5.018 |

Table 2: Quantitative UIQM values of samples in Figure 6. The greater the UIQM values, the better the enhanced results, with blue representing the maximum and green representing the minimum.

| Assessments | Methods | (a) | (b) | (c) | (d) | (e) | (f) | (g) | (h) | (i) | (j) |
|---|---|---|---|---|---|---|---|---|---|---|---|
| | Input | 4.865 | 4.316 | 4.923 | 4.516 | 3.854 | 4.837 | 3.740 | 4.320 | 4.819 | 3.468 |
| | UCM | 5.127 | 5.093 | 4.999 | 5.165 | 4.558 | 5.056 | 4.028 | 4.925 | 4.773 | 4.634 |
| | HE | 4.942 | 5.189 | 5.320 | 5.016 | 4.809 | 5.153 | 4.608 | 4.910 | 4.892 | 4.618 |
| | MSRCR | 4.747 | 4.700 | 4.096 | 4.908 | 4.426 | 4.039 | 3.906 | 4.067 | 3.606 | 4.318 |
| UIQM | Fusion | 5.280 | 5.131 | 5.063 | 5.184 | 4.485 | 5.030 | 4.023 | 4.811 | 4.961 | 4.557 |
| | UDCP | 5.389 | 5.256 | 4.932 | 4.868 | 4.731 | 5.406 | 4.722 | 5.180 | 4.962 | 5.131 |
| | IBLA | 5.158 | 4.796 | 4.560 | 4.626 | 3.978 | 4.858 | 3.873 | 4.494 | 3.965 | 4.139 |
| | UGAN | 5.249 | 5.185 | 5.040 | 4.800 | 4.832 | 5.026 | 4.561 | 5.601 | 4.934 | 4.675 |
| | WaterGAN | 5.003 | 4.537 | 4.756 | 4.636 | 4.212 | 4.524 | 3.801 | 4.323 | 4.846 | 4.223 |
| | Ours | 5.391 | 5.058 | 4.979 | 4.891 | 4.834 | 5.015 | 4.034 | 4.936 | 5.140 | 4.377 |

Table 3: Average quantitative UICM, UISM, UIConM and UIQM values on real-world underwater image datasets RealA, RealB and RealC. The greater the values, the better the enhanced results, with blue representing the maximum

| Datasets | Assessments | Input | UCM | HE | MSRCR | Fusion | UDCP | IBLA | UGAN | WaterGAN | Ours |
|---|---|---|---|---|---|---|---|---|---|---|---|
| | UICM | -0.332 | -0.059 | 0.003 | -0.006 | -0.127 | -0.300 | -0.233 | -0.074 | -0.079 | 0.006 |
| RealA | UISM | 7.151 | 7.092 | 7.194 | 6.934 | 7.000 | 7.073 | 7.148 | 7.045 | 6.820 | 7.096 |
| | UIConM | 0.593 | 0.694 | 0.812 | 0.537 | 0.716 | 0.739 | 0.679 | 0.770 | 0.634 | 0.675 |
| | UIQM | 4.22 | 4.574 | 5.027 | 3.967 | 4.622 | 4.721 | 4.533 | 4.832 | 4.280 | 4.508 |
| | UICM | -0.273 | 0.029 | 0.016 | -0.006 | -0.0350 | -0.051 | -0.193 | -0.120 | -0.151 | 0.091 |
| RealB | UISM | 7.169 | 7.053 | 7.120 | 6.944 | 6.910 | 7.080 | 7.049 | 6.957 | 6.821 | 6.992 |
| | UIConM | 0.506 | 0.730 | 0.772 | 0.654 | 0.737 | 0.837 | 0.703 | 0.804 | 0.643 | 0.708 |
| | UIQM | 3.920 | 4.695 | 4.864 | 4.387 | 4.675 | 5.080 | 4.590 | 4.927 | 4.309 | 4.598 |
| | UICM | -0.223 | -0.023 | -0.010 | 0.006 | -0.110 | -0.085 | -0.136 | -0.089 | -0.121 | 0.044 |
| RealC | UISM | 7.310 | 7.309 | 7.312 | 7.348 | 7.318 | 7.428 | 7.305 | 7.117 | 6.895 | 7.282 |
| | UIConM | 0.674 | 0.740 | 0.743 | 0.493 | 0.764 | 0.964 | 1.207 | 0.810 | 0.824 | 0.780 |
| | UIQM | 4.561 | 4.803 | 4.816 | 3.932 | 4.891 | 5.636 | 6.469 | 4.996 | 4.979 | 4.942 |

UIQM is a non-reference assessment metric whose quantitative results depend largely on the value of scale factors. Structural information of images is not considered in these kinds of non-reference evaluation metrics. Although some enhanced images can get higher score, the visual quality is poor, the reason is that the metric is calculated from the pixels. Therefore, we also employ three full-reference assessment metrics MSE, PSNR, and SSIM to evaluate the performance of different methods on synthetic datasets without training. The comparison results in Table 4 demonstrate that our proposed method achieves the best results in terms of MSE, PSNR, and SSIM.

The average inference time of different algorithms are compared in one computer with following configuration: Intel i7-8750H CPU, 16GB RAM, and GTX1060 6G GPU. The results are shown in Table 5. Our model has the fastest processing speed compared to other methods. Moreover, the model we proposed has the fewest Params and FLOPs compared to other deep-learning-based methods. UGAN employs many convolution layers with 512 kernels, which causes that there are too many network parameters. WaterGAN employs multiple networks, resulting in slow processing speed.

Table 4: Quantitative results evaluation on synthetic dataset by full-reference metrics: MSE, PSNR, SSIM values. The smaller the MSE values, the greater the PSNR and SSIM values, the better the enhanced results, with blue representing the best results

| Datasets | Assessments | Input | UCM | HE | MSRCR | Fusion | UDCP | IBLA | UGAN | WaterGAN | Ours |
|---|---|---|---|---|---|---|---|---|---|---|---|
| | MSE | 0.042 | 0.029 | 0.045 | 0.059 | 0.027 | 0.072 | 0.058 | 0.026 | 0.014 | 0.002 |
| Synthesis | PSNR | 20.68 | 23.46 | 18.315 | 13.25 | 23.13 | 17.37 | 19.10 | 20.63 | 20.25 | 30.31 |
| | SSIM | 0.869 | 0.944 | 0.845 | 0.580 | 0.933 | 0.847 | 0.832 | 0.779 | 0.842 | 0.966 |

Table 5: Testing time and parameters of generator of different enhancement methods

| | UCM | HE | MSRCR | Fusion | UDCP | IBLA | UGAN | WaterGAN | Ours |
|---|---|---|---|---|---|---|---|---|---|
| Testing time (s) | 1.284 | 0.009 | 0.076 | 0.118 | 2.051 | 4.561 | 0.022 | 10.347 | 0.008 |
| Params (M) | - | - | - | - | - | - | 54.41 | 28.62 | 1.93 |
| FLOPs (M) | - | - | - | - | - | - | 610 | 8053 | 3.8 |

As indicated by some previous works (Uplavikar P, 2019; Anwar S, 2019; Ding X, 2019), the performance of high-level computer vision tasks (such as underwater target detection) on enhanced images is an indicator of image enhancement methods. We applied YOLO v3 (Redmon J, 2018) target detector on degraded underwater images and their enhanced versions generated by our model. The performance of underwater target detection is better on enhanced versions on degraded images. Figure 7 shows the results of YOLO v3 detector before and after processing the images with our model.

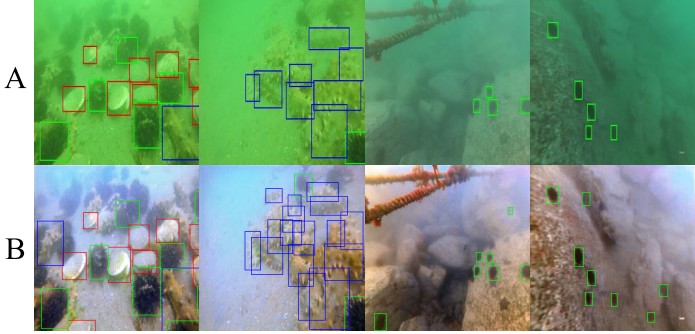

Figure 7: Underwater target detection results before and after enhancement. (A) Real-world underwater images and (B) output of our model for the real-world image. Red boxes represent scallops, blue boxes represent sea cucumbers, and green boxes represent sea urchins.

## 5 CONCLUSION

Based on an improved underwater imaging model, a generative adversarial network (UWGAN) for generating realistic underwater images is proposed in this paper. Then, U-net with combined loss functions is used for degraded underwater images enhancement. Our model is validated on both low-level and high-level underwater computer vision tasks, which demonstrate its effectiveness and robustness.

## ACKNOWLEDGEMENT

The authors would like to acknowledge the National Key R&D Program of China (Grant No. 2018YFC0309402) for funding this work.

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

## A  LOSS FUNCTIONS

The most common loss function for image restoration is L2 error. However, which loss function is suitable for underwater image enhancement has not been studied. Inspired by a related article, the effect of different loss functions in U-net is studied in this paper. Table 6 shows the loss functions we used.

In mathematical formula, $x$ is an index of pixels in region $X$, $g(x)$ is pixel value in region $X$ of the image reconstructed by U-net and $r(x)$ is the pixel value of corresponding ground truth. $\overline{x}$ is the central pixel value of region $X$. $\nabla g(x)$, $\nabla r(x)$ respectively represent the gradient of reconstructed images and clear images. After several experiments and observations of the best reconstruction results, we set $\alpha$ to 0.8 in this paper.

Table 6: Different loss functions for underwater image restoration. Including some basic loss functions and their combinations.

| Name | Mathematical formula |
|---|---|
| The $L1$ loss error | $\mathcal{L}^{l1}(X) = \dfrac{1}{N}\sum\limits_{x \in X} \lvert g(x) - r(x) \rvert$ |
| The $L2$ loss error | $\mathcal{L}^{l2}(X) = \dfrac{1}{N}\sum\limits_{x \in X} \left(g(x) - r(x)\right)^2$ |
| The $SSIM$ loss error | $\mathcal{L}^{SSIM}(X) = \dfrac{1}{N}\sum\limits_{x \in X} 1 - SSIM(x)$ |
| The $MSSSIM$ loss error | $\mathcal{L}^{MS-SSIM}(X) = 1 - MS\_SSIM(\overline{x})$ |
| The $GDL$ error | $\mathcal{L}^{gdl}(X) = \dfrac{1}{N}\sum\limits_{x \in X} \lvert \nabla g(x) - \nabla r(x) \rvert$ |
| $L1 + L2$ | $\mathcal{L}^{l1\_l2}(X) = \alpha \cdot \mathcal{L}^{l2} + (1 - \alpha) \cdot \mathcal{L}^{l1}$ |
| $L1 + SSIM$ | $\mathcal{L}^{l1\_SSIM}(X) = \alpha \cdot \mathcal{L}^{SSIM} + (1 - \alpha) \cdot \mathcal{L}^{l1}$ |
| $L1 + MSSSIM$ | $\mathcal{L}^{l1\_MS-SSIM}(X) = \alpha \cdot \mathcal{L}^{MS-SSIM} + (1 - \alpha) \cdot \mathcal{L}^{l1}$ |
| $L1 + GDL$ | $\mathcal{L}^{l1\_gdl}(X) = \alpha \cdot \mathcal{L}^{gdl} + (1 - \alpha) \cdot \mathcal{L}^{l1}$ |

## B  ABLATION STUDY

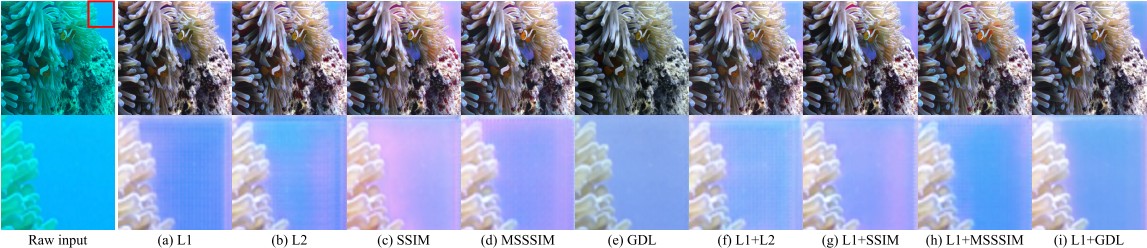

Raw input  (a) L1  (b) L2  (c) SSIM  (d) MSSSIM  (e) GDL  (f) L1+L2  (g) L1+SSIM  (h) L1+MSSSIM  (i) L1+GDL

Figure 8: The visual quality of the sample image in RealC dataset with different loss functions. From (a) to (i) are respectively enhanced results of the loss function $L1$, $L2$, $SSIM$, $MSSSIM$, $GDL$, $L1 + L2$, $L1 + SSIM$, $L1 + MSSIM$, and $L1 + GDL$.

Ablation study is mainly to reveal the effects of different loss functions. We use different loss functions to train the network and test it on RealC, and synthetic datasets. The sample image is selected from the RealC dataset, as shown in Figure 8, the enhanced results range from (a)~(i) are obtained with different loss functions, and images in the second row show the details in the red box area of the image. It can be seen from

the results in the second row, (a), (b), (f) appear striped artifacts. (c), (d), (g) cause color unevenness. The details of (e) are natural but it lacks sufficient saturation. (h), (i) show proper enhanced results, the color in (h) is more vivid but with slightly striped artifacts.

The enhanced result using the $L1$ or $L2$ loss function appears stripe-like artifacts while the $SSIM$ or $MSSSIM$ loss function causes color unevenness. The enhanced result of the $GDL$ loss function is natural but lacks sufficient saturation. We calculated the MSE, PSNR, and SSIM metrics on the synthetic dataset. The quantitative scores in Table 7 demonstrate that a combination of multiple loss functions can achieve better enhancement results.

Table 7: Quantitative results of different loss functions evaluation on synthetic dataset by full-reference metrics: MSE, PSNR, SSIM values.

| Datasets | Assessments | Input | L1 | L2 | SSIM | MSSSIM | GDL | L1+L2 | L1+SSIM | Ll+MSSSIM | L1+GDL |
|---|---|---|---|---|---|---|---|---|---|---|---|
| | MSE | 0.0417 | 0.002 | 0.002 | 0.002 | 0.002 | 0.009 | 0.001 | 0.001 | 0.002 | 0.002 |
| Synthesis | PSNR | 20.68 | 29.91 | 28.72 | 29.99 | 28.27 | 25.21 | 32.97 | 32.82 | 30.31 | 30.81 |
| | SSIM | 0.867 | 0.962 | 0.959 | 0.974 | 0.960 | 0.944 | 0.971 | 0.979 | 0.966 | 0.968 |

