# OpenReview forum: "UWGAN: UNDERWATER GAN FOR REAL-WORLD UNDERWATER COLOR RESTORATION AND DEHAZING"
_ICLR.cc/2020/Conference — Reject_

### Official Review · AnonReviewer1 · 2019-10-23
**Official Blind Review #1**

**Rating:** 3

**Review:**

[Update after rebuttal period]
In response, the authors cannot clearly clarify the difference between this work with existing works integrating the physical model into the network. Thus I stay my original score.


[Original reviews]
This paper proposed an unsupervised generative adversarial network for underwater generating realistic underwater images and haze removal, which can simultaneously deal with the color restoration and haze in the realistic underwater environment.

Firstly, according to the widely used physical model in the image processing area, employed the UnderwaterGAN to trained parameters in advanced, and then use U-Net for color restoration and haze removal of underwater images. However, many existing works used the physical model to represent the imaging principles and using deep network to learn prior knowledge. Thus, I think the proposed idea is a little bit incremental.

For the experimental part, the experimental results fully demonstrate the effectiveness of the proposed method in comparison with state-of-the-art methods. Additionally, the ablation studies in the appendix also give us the intuition by using the different loss functions. Also, I suggest the authors demonstrate the proposed method on not only low-level, but also high-level vision tasks, e.g., underwater image target detection.

Finally, the paper is well organized and sentence expression is also clear, but small errors that are correctable.


**Experience Assessment:**

I have published in this field for several years.

**Review Assessment: Checking Correctness Of Derivations And Theory:**

N/A

**Review Assessment: Checking Correctness Of Experiments:**

I assessed the sensibility of the experiments.

**Review Assessment: Thoroughness In Paper Reading:**

I read the paper at least twice and used my best judgement in assessing the paper.

---

> ### Author Response · Authors · 2019-11-13
> **Response**
>
> We would like to thank the reviewer for pointing out some problems in our work. Please find our response to your questions below. We have updated the paper and uploaded a revision on Nov 13.
>
> **1) Many existing works used the physical model to represent the imaging principles and using deep network to learn prior knowledge**
>
> **Response:** Inspired by in-air images dehazing algorithms, we improved the underwater imaging model in this paper, but it has not been stated clearly before. Then, we employed GAN for generating more realistic underwater-style images based on the improved model (taken both light attenuation and haze effect in real-world underwater images into consideration), which can be found in Sections 2.1 and 2.2 in this paper.
>
> We employed U-Net as an enhancement network structure, but not only that, we studied the effect of different loss functions in U-Net (The detailed content can be found in Page 11, section APPENDIX), which could provide a new idea for further research about loss functions on underwater image enhancement. Considering the inference speed and Flops, U-Net is better than other networks and could run in real-time compared to other deep-learning-based methods mentioned in this paper.
>
> **2) High-level vision tasks **
>
> **Response:** We applied YOLO v3 target detector on degraded underwater images and their enhanced versions generated by our model. The performance of underwater target detection is better on enhanced versions of degraded images，which demonstrated our proposed method on high-level underwater computer-vision tasks. This part can be found in section 4, Page 8.
>
> **3) Small errors**
>
> **Response:** We have modified some sentence expressions and small errors in this paper.

---

### Official Review · AnonReviewer2 · 2019-10-23
**Official Blind Review #2**

**Rating:** 3

**Review:**

This paper uses U-net for underwater image restoration and enhancement. But, it is difficult to obtain realistic underwater images, thus this paper introduces a GAN-based method to generate realistic underwater images from in-air image and depth map pairs.

- Although this paper points out that the previous work (i.e. WaterGAN) generates color noise and the camera model is not suitable, how does this proposed method overcome these points? Please make it clear.

- The figures in this paper are too blurry to see them. To evaluate the effectiveness of the proposed method, the figures are important, thus, it would be better to make them clear.

- The technical contribution of the proposed method is not clear. The proposed method seems to be just using the existing techniques.

**Experience Assessment:**

I have read many papers in this area.

**Review Assessment: Checking Correctness Of Derivations And Theory:**

N/A

**Review Assessment: Checking Correctness Of Experiments:**

I assessed the sensibility of the experiments.

**Review Assessment: Thoroughness In Paper Reading:**

I read the paper at least twice and used my best judgement in assessing the paper.

---

> ### Author Response · Authors · 2019-11-13
> **Response**
>
> We would like to thank the reviewer for pointing out some problems in our work. Please find our response to your questions below. We have updated the paper and uploaded a revision on Nov 13.
>
> **1) This paper points out that the previous work (i.e. WaterGAN) generates color noise and the camera model is not suitable, how does this proposed method overcome these points?**
>
> **Response:** The network structure of WaterGAN can be found here (https://github.com/kskin/WaterGAN).
>
> The image synthesized by WaterGAN suffers color noise due to the input of noise vector z, which was observed when we tested WaterGAN.
>
> “One limitation of our model is in the parameterization of the vignetting model, which assumes a centered vignetting pattern. This is not a valid assumption for the MHL dataset, so our restored images still show some vignetting though it is partially corrected.”, This sentence is mentioned in the original paper of WaterGAN (In page 6, above VI. Conclusion). It is our mistake to call vignetting model as camera model in Introduction part.
>
> In order to solve the color noise problem, noise vector z is no longer necessary in our UWGAN model, UWGAN takes color image and its depth map as input, so our model can avoid color noise problem. And, we didn't use “vignetting model” in our model. Inspired by in-air images dehazing algorithms, we improved the underwater imaging model in this paper. Then, we employed GAN for generating more realistic underwater-style images based on the improved model (taken both light attenuation and haze effect in real-world underwater images into consideration), which can be found in Sections 2.1 and 2.2 in this paper.
>
> **2) The figures in this paper are too blurry to see them**
>
> **Response:** We have improved the resolution of images in this paper. It should support higher magnifications.
>
> **3) The technical contribution of the proposed method is not clear**
>
> **Response:** As mentioned in the first paragraph, inspired by in-air images dehazing algorithms, we improved the underwater imaging model in this paper. Then, we employed GAN for generating more realistic underwater-style images based on the improved model (taken both light attenuation and haze effect in real-world underwater images into consideration), which can be found in Sections 2.1 and 2.2 in this paper. We employed U-Net as an enhancement network structure, but not only that, we studied the effect of different loss functions in U-Net (The detailed content can be found in Page 11, section APPENDIX), which could provide a new idea for further research about loss functions on underwater image enhancement. Considering the inference speed and Flops, U-Net is better than other networks and could run on real-time compared to other deep-learning-based methods mentioned in this paper.

---

### Official Review · AnonReviewer3 · 2019-11-06
**Official Blind Review #3**

**Rating:** 3

**Review:**

In this article, the authors propose a generative adversarial network named UWGAN to generate realistic underwater images from the pairs of in-air images and depth images. Then, a U-Net was leveraged to enhance the results.
However, the text suffers from too many language problems. The authors should consult professional proofreading services. As a courtesy towards referees, the quality of writing needs meticulous attention before a scientific paper should be submitted.
	Other comments:
1.	The literature is limited. I found some novel works being done in the field that must be addressed and listed in the background and experiments.
2.	The underwater imaging model presented in this paper derives from the Jaffe-McGlamery model, which is a common sense in this field. The authors use a generator to produce underwater images that only implements the common model by a neural network. Moreover, the statement of section 2.2 is not clear. Please rewrite this section.
3.	The authors used U-Net without any improvement to enhance the results generated from UWGAN, which is the integration of existing models.
4.	The authors claimed that their model is better than others, while there is no evidence to indicates that. For example, 1) in (page 5, line 4 from bottom), “It can be seen that our proposed method has achieved a higher score.”, can we observe this from the Table 1 and 2? 2) “The method we proposed has the fastest processing speed compared to other methods. Moreover, the method proposed in this paper has the fewest parameters compared to other deep-learning-based methods.”, it is suggested that a study about the parameters and FLOPs of the involved methods should be given.
5.	Please carefully check the references. For example, “Hummel R. Image enhancement by histogram transformation[J]. Unknown, 1975.” lacks the journal name.
6.	High-resolution figures should be given in the manuscript.


**Experience Assessment:**

I have published one or two papers in this area.

**Review Assessment: Checking Correctness Of Derivations And Theory:**

I assessed the sensibility of the derivations and theory.

**Review Assessment: Checking Correctness Of Experiments:**

I assessed the sensibility of the experiments.

**Review Assessment: Thoroughness In Paper Reading:**

I read the paper thoroughly.

---

> ### Author Response · Authors · 2019-11-13
> **Response**
>
> We would like to thank the reviewer for pointing out some problems in our work. Please find our response to your questions below. We have updated the paper and uploaded a revision on Nov 13.
>
> **1) The literature is limited.**
>
> **Response:** We have cited and listed some new references in the background. In section 4, the method we chose to compare with ours can be roughly divided into three types: model-free algorithms, model-based algorithms, and deep-learning-based algorithms. These methods are classical and representative, we can't list too much due to paper length limit.
>
> > Anwar S, Li C, Porikli F. Deep underwater image enhancement[J]. arXiv preprint arXiv:1807.03528, 2018.
> >
> > Ancuti C, Ancuti C O, De Vleeschouwer C. D-hazy: A dataset to evaluate quantitatively dehazing algorithms[C]//2016 IEEE International Conference on Image Processing (ICIP). IEEE, 2016: 2226-2230.
> >
> > Uplavikar P, Wu Z, Wang Z. All-In-One Underwater Image Enhancement using Domain-Adversarial Learning[J]. arXiv preprint arXiv:1905.13342, 2019.
> >
> > Anwar S, Li C. Diving Deeper into Underwater Image Enhancement: A Survey[J]. arXiv preprint arXiv:1907.07863, 2019.
> >
> > Ding X, Wang Y, Yan Y, et al. Jointly Adversarial Network to Wavelength Compensation and Dehazing of Underwater Images[J]. arXiv preprint arXiv:1907.05595, 2019.
> >
> > Redmon J, Farhadi A. Yolov3: An incremental improvement[J]. arXiv preprint arXiv:1804.02767, 2018.
>
> **2) The underwater imaging model presented in this paper derives from the Jaffe-McGlamery model, which is a common sense in this field. The authors use a generator to produce underwater images that only implements the common model by a neural network. Moreover, the statement of section 2.2 is not clear. Please rewrite this section.**
>
> **Response:** We have rewritten section 2.1 and 2.2. Inspired by in-air images dehazing algorithms, we improved the underwater imaging model in this paper. Then, we employed GAN for generating more realistic underwater-style images based on the improved model (taken both light attenuation and haze effect in real-world underwater images into consideration), which can be found in Sections 2.1 and 2.2 in this paper.
>
> **3) The authors used U-Net without any improvement to enhance the results generated from UWGAN, which is the integration of existing models.**
>
> **Response:** U-Net is an efficient tool for the proposed pipeline. We employed U-Net as an enhancement network structure, but not only that, we studied the effect of different loss functions in U-Net (The detailed content can be found in Page 11, section APPENDIX), which could provide a new idea for further research about loss functions on underwater image enhancement. Considering the inference speed and Flops, U-Net is better than other networks and could run on real-time compared to other deep-learning-based methods mentioned in this paper.
>
> **4) The authors claimed that their model is better than others, while there is no evidence to indicates that.**
>
> **Response:** We have revised some imprecise sentences of the result analysis part in section 4, “Table 1 and Table 2 quantitatively show the scores of sample images in Figure 5 and Figure 6 respectively. It can be seen that our proposed method has achieved the highest scores in (a), (c) and (f). In addition, the average quantized scores evaluated on RealA, RealB, and RealC datasets are shown in Table 3. Our model achieves the best score in terms of color restoration.” Besides, we add FLOPs of deep-learning-based methods in Table 5.
>
> **5) Please carefully check the references.**
>
> **Response:** We have revised small errors in references.
>
> > “Hummel R. Image enhancement by histogram transformation[J]. Computer Graphics and Image Processing, 1977, 6(2):184-195.”
>
> **6) High-resolution figures should be given in the manuscript.**
>
> **Response:**  We have improved the resolution of images in this paper. It should support higher magnifications.

---

### Public Comment · ~Chenxu_John_Wang1 · 2019-11-03
**Author names are shown in paper**

Ain't this a violation of double blind reviewing policy?

---

> ### Author Response · Authors · 2019-11-03
> **Reply to "author names are shown in paper"**
>
> This is our first-time submission to ICLR. I am very sorry to have made a mistake. Is there anything we can do to correct this mistake? Can we resubmit our paper with hiding the authors' names?

---

### Author Response · Authors · 2019-11-13
**Summary of Revision**

Following reviewers' suggestions, we have updated the paper and uploaded a revision on Nov 13. Here we give a summary of the major changes.

1. We have rewritten section 2.1 and section 2.2. Now we clearly state the improved underwater imaging model and our technical approaches.
2. In section 4, we add the results of underwater target detection,  which demonstrates that our proposed model can help in underwater high-level computer vision tasks.
3. We have replaced all figures with higher resolution versions in this paper.
4. We add FLOPs in Table 5.
5. We fixed small errors and inappropriate sentence expression.

---

### Decision · Program_Chairs · 2019-12-19

**Decision:**

Reject

**Comment:**

This paper proposed to improve the quality of underwater images, specifically color distortion and haze effect, by an unsupervised generative adversarial network (GAN). An end-to-end autoencoder network is used to demonstrate its effectiveness in comparing to existing works, while maintaining scene content structural similarity. Three reviewers unanimously rated weak rejection. The major concerns include unclear difference with respect to the existing works, incremental contribution, low quality of figures, low quality of writing, etc. The authors respond to Reviewers’ concerns but did not change the rating. The ACs concur the concerns and the paper can not be accepted at its current state.